# Role of Maternal Antibodies in the Protection of Broiler Chicks against *Campylobacter* Colonization in the First Weeks of Life

**DOI:** 10.3390/ani14091291

**Published:** 2024-04-25

**Authors:** Kristof Haems, Diederik Strubbe, Nathalie Van Rysselberghe, Geertrui Rasschaert, An Martel, Frank Pasmans, An Garmyn

**Affiliations:** 1Department of Pathobiology, Pharmacology and Zoological Medicine, Ghent University, B9820 Merelbeke, Belgium; 2Terrestrial Ecology Unit (TEREC), Ghent University, B9000 Ghent, Belgium; 3Technology & Food Sciences Unit, Flanders Research Institute for Agriculture, Fisheries and Food (ILVO), B9090 Melle, Belgium

**Keywords:** *Campylobacter*, maternal antibody, broiler, broiler breeder, vaccination

## Abstract

**Simple Summary:**

*Campylobacter* is the main cause of foodborne human gastroenteritis worldwide. Most of these cases originate from the consumption of poultry products. Broiler chickens are often colonized by this bacterium, with a high prevalence at slaughter age. However, in the first 2–3 weeks of life, chicks are resistant to *Campylobacter* infection, and this resistance has been attributed to the transfer of maternal antibodies from the hen’s serum to the yolk and then to the hatched chick’s bloodstream and intestines. In this study, the role of maternal antibodies in the protection of broilers against *Campylobacter* colonization was investigated. Field monitoring of broiler flocks from breeders with varying antibody levels showed a trend of lower *Campylobacter* prevalence in offspring from high-antibody-level breeders. In a series of trials, breeders were vaccinated to increase maternal antibody levels in offspring, with the aim of increasing the chicks’ resistance to infection. A minor reduction in *Campylobacter* prevalence was obtained only in the second week of life. Immunization of breeders thus showed limited efficacy in enhancing protection against *Campylobacter* infection in the offspring.

**Abstract:**

Thermophilic *Campylobacter* species are the most common cause of bacterium-mediated diarrheal disease in humans globally. Poultry is considered the most important reservoir of human campylobacteriosis, but so far, no effective countermeasures are in place to prevent the bacterium from colonizing broiler flocks. This study investigated maternal antibodies’ potential to offer protection against *Campylobacter* in broiler chicks via a field trial and an immunization trial. In the field trial, breeder flocks with high and low anti-*Campylobacter* antibody levels in the yolk were selected based on serological screening. Offspring were subsequently monitored for maternal antibodies and *Campylobacter* prevalence during early life. Although maternal antibodies declined rapidly in the serum of broilers, offspring from flocks with lower anti-*Campylobacter* antibody levels seemed to be more susceptible to colonization. In the immunization trial, breeders from a seropositive breeder flock were vaccinated with an experimental bacterin or subunit vaccine. Immunization increased antibody levels in the yolk and consequently in the offspring. Elevated maternal antibody levels were significantly associated with reduced *Campylobacter* susceptibility in broilers at 2 weeks old but not at 1 and 3 weeks old. Overall, the protective effect of maternal immunity should be cautiously considered in the context of *Campylobacter* control in broilers. Immunization of breeders may enhance resistance but is not a comprehensive solution.

## 1. Introduction

*Campylobacter*, a genus of Gram-negative bacteria, is the leading cause of human gastroenteritis worldwide. The most common symptoms of campylobacteriosis include fever, diarrhea and abdominal cramps. Although most of these infections are self-limiting, they can lead to more serious pathologies such as extraintestinal diseases and chronic sequelae like reactive arthritis or Guillain-Barré syndrome [1]. In 2022, 137,107 confirmed cases of campylobacteriosis were reported in 27 EU countries, with an overall incidence rate of 43.1 cases per 100,000. The majority (87.6%) of these infections were caused by *Campylobacter (C.) jejuni*, followed by *C. coli* (10.7%) [2].

The transmission of the bacteria to humans typically occurs through direct or indirect contact with animal-derived products, particularly poultry products. In Belgium, 30 to 70% of broiler flocks are positive for *Campylobacter* at slaughter age [3,4,5,6]. Thermophilic *Campylobacter* species colonize the ceca of chickens to a high degree (10^6^–10^9^ colony forming units (cfu)/g) and are often regarded as a commensal. However, colonization of broiler chickens can be associated with intestinal inflammation, leading to diarrhea and subsequent associated pathology [7]. More likely, the interaction with the chicken host suggests that their relationship can better be described as “immunological tolerance”, as *Campylobacter* may escape or alter the inflammatory response via different mechanisms [8].

Currently, there are no effective counter measures in place to prevent the colonization of broilers at the farm level [9,10]. *Campylobacter* is typically introduced horizontally in the chicken flock through different environmental sources and various vectors such as rodents and insects [11]. Once present in the poultry house, the bacteria spread quickly, and one week after initial colonization, most chickens in the flock are already colonized [12]. Although it has been suggested [13] that fecal contamination of the eggshell can lead to the possible survival of *Campylobacter* in the eggshell membranes and transmission to the progeny, other authors claim that the transfer of *Campylobacter* from parent stock to the offspring is unlikely in commercial settings [14,15]. Moreover, *Campylobacter* is rarely detected in chicks under 2 to 3 weeks of age [16,17], the time before which is referred to as the lag period. This lag could be attributed to competition among microflora, which inhibits proliferation [18], as well as to the presence of maternal antibodies, which decline by about 2 weeks of age [19,20]. These antibodies contribute to the complement-mediated killing of *C. jejuni* in a strain-specific manner [20,21]. The immunization of specific-pathogen-free (SPF) broiler breeders with a bacterin and subunit vaccine led to the protection of their offspring against *Campylobacter* colonization during the first two to three weeks of their life due the transfer of maternal antibodies [22].

This study seeks to further investigate the potential protective role of maternal antibodies against *Campylobacter* in chicks. As there is a lack of field studies investigating the impact of anti-*Campylobacter* maternal antibodies, this research first aimed to examine the link between the presence of anti-*Campylobacter* antibodies in broiler breeders and the onset of *Campylobacter* colonization in their offspring. Then, we immunized seropositive breeders using two experimental vaccines to assess the extent to which resulting anti-*Campylobacter* antibody levels might improve protection of their progeny against *Campylobacter* colonization.

## 2. Materials and Methods

### 2.1. Field Study: Anti-Campylobacter IgY in Broiler Breeders’ Egg Yolks and Resulting IgY Titers and Probability of Colonization in Their Offspring

A total of 20 eggs per flock were obtained from 25 different broiler breeder flocks via a local hatchery (Belgabroed NV, Merksplas, Belgium), with varying ages ranging from 27 to 44 weeks. For quantification of the IgY titers, the egg yolks were analyzed via ELISA, as described below (Section 2.7). At first, the eggs were pooled by flock. Based on these results, flocks were assigned to one of two groups: breeder flocks with high anti-*Campyloacter* IgY titers and breeder flocks with low anti-*Campylobacter* IgY titers. The cut-off values for the group with low and high antibody titers were anti-*Campylobacter* IgY titers ≤ 1:1600 and ≥1:3200 respectively. Subsequently, all the eggs from the selected flocks were analyzed individually. Flocks from the group with low IgY titers that differed significantly in optical density 450 nm (OD450) from the flocks in the group with high IgY titers were selected for follow-up of the progeny flocks and vice versa. Pooled cecal samples from the selected flocks were collected for *Campylobacter* culture, as described below (Section 2.4).

Following selection, 20 broiler flocks descended from the selected breeder flocks were monitored. These flocks included 10 flocks from the group with low anti-*Campylobacter* IgY titers and 10 flocks from the group with high anti-*Campylobacter* IgY titers. Fifteen randomly selected broiler chicks per flock were monitored at weekly intervals, with monitoring starting at 7 days of age and ending at 28 days of age. A fixed volume of 1 mL of serum was collected from each chick, and the collected serum from all chicks on the same farm was pooled and mixed weekly. The pooled samples were subsequently analyzed for maternal antibodies via ELISA, following the procedure described below (Section 2.7). In anticipation of processing, the samples were stored at −20 °C. Next, the chicks were euthanized by injection of an overdose (100 mg/kg) of sodium pentobarbital (Kela, Elversele, Belgium) in the wing vein, and the cecal content was collected for detection of *Campylobacter*, as described below (Section 2.4). For a schematic overview, see Figure 1.

### 2.2. Vaccination Study: Selection and Immunization of in Field Seropositive Broiler Breeders and Determination of Maternal Antibodies after Vaccination

To select the breeder flock for the vaccination trial, eggs (*n* = 10/flock) from 4 different broiler breeder flocks with ages between 23 and 26 weeks were collected. These eggs were acquired from the same local hatchery as sampled previously (Belgabroed NV, Merksplas, Belgium). The anti-*Campylobacter* IgY titers in the yolk of individual eggs were analyzed via ELISA as described below (Section 2.7). The flock with the highest titer (1:2000 ± 400) and a low coefficient of variation (CV%) (63%) was selected for vaccination purposes. On the day of sampling, this flock was 26 weeks old. The other breeder flocks had titers of 1:180 ± 47, 1:920 ± 298 and 1:840 ± 139. From the selected flock, 45 commercial Ross 308 broiler breeder hens and 10 Ross 308 broiler breeder roosters were purchased. On arrival, pooled cecal droppings were collected for *Campylobacter* culture, as described below (Section 2.4). Flocks were subjected to *Campylobacter* screening to characterize the colonizing strains. This investigation aimed to determine whether the antibodies circulating in the flock, which originated from field infections, were homologous or heterologous to the challenge strain utilized for offspring inoculation. The goal was to assess the efficacy of the maternal antibodies in providing protection in the offspring of sham-immunized negative-control breeders. Hens were randomly assigned to three vaccine groups, and each group was separated into three subgroups: bacterin (*n* = 15), subunit (*n* = 15) and sham-vaccinated negative control (*n* = 15), in a design similar to that used in the preceding study [22]. Birds were housed in a minimum enclosure space of 0.21 m^2^ per bird. The birds were given restricted commercial food (Versele Laga, Deinze, Belgium) once per day in the morning. The feeding schedule and light schedule were adapted based on the Ross Broiler Breeder Management Guide and the conditions on the originating farm. Drinking water was provided ad libitum.

At 30 weeks of age, breeders were primo-vaccinated. This vaccination was followed by 3 boosters at two-week intervals, in a design similar to that used in the previous study [22]. The vaccine was administered by intramuscular injection in the pectoral muscle. The composition of the vaccines is described below (Section 2.6). Two weeks after each vaccination and 2 and 4 months after the final booster, yolk and blood derived from the wing vein was collected and pooled by group (control, bacterin and subunit). In anticipation of processing, the samples were stored at −20 °C. The anti-*Campylobacter* IgY titers were determined via ELISA, as described below (Section 2.7). Two weeks and two months after the last booster, the first fertilized eggs from these (sham-)vaccinated broiler breeders were collected and incubated. Post-hatch, the progeny were used to determine the probability of colonization with *Campylobacter* in threshold trials, as described below (Section 2.3). Husbandry, euthanasia methods, experimental procedures and biosafety precautions were approved by the Ethical Committee (EC2021_086) of the Faculty of Veterinary Medicine, Ghent University, Belgium.

### 2.3. Vaccination Study: Protective Efficacy of Bacterin and Subunit Derived Maternal Antibodies on Cecal Colonization in Broilers Using a Threshold Model

The susceptibility of young chicks derived from immunized field breeders to initial colonization by *Campylobacter* was studied using a threshold model [22,23]. Chicks were housed individually to prevent the spread of *Campylobacter* to other chicks so that the individual probability of a chick becoming colonized by a fixed dose of *Campylobacter* could be assessed. Fertilized eggs from the three (sham-)vaccinated groups (bacterin, subunit and control) were hatched and housed in groups until the age of inoculation. Birds were housed according to the guidelines set by the ethical committee, and conditions were similar to those used in the preceding study [22]. This means that they were raised on a bedding of wood shavings and under a heat lamp. The minimum enclosure space depended on the weight of the birds (from 0.025 m^2^ per bird at week 1 to 0.09 m^2^ at week 3). From the day of hatching until the end of the experiment, chicks received ad libitum commercial starting feed (Farm 1 & 2, Versele Laga, Belgium) and drinking water. From different time-points onwards (7, 14 and 21 days of age), each chick was housed individually in a box on a bedding of wood shavings with an enclosure space of 0.16 m^2^ per chick and audiovisual contact with the other chicks. Before chicks were challenged, cecal samples were collected, pooled by group and tested for *Campylobacter* isolation. Chicks were orally inoculated in the crop with *Campylobacter jejuni* strain KC40 [22,23,24,25,26] using a syringe, administering 1 mL inoculum containing either a low dose (2.7–3.0 log_10_ cfu) or a high dose (3.6–4.0 log_10_ cfu). Two days after inoculation, the chicks were euthanized by injection of an overdose (100 mg/kg) of sodium pentobarbital (Kela, Hoogstraten, Belgium) in the wing vein and the cecal content was collected for qPCR analysis, as described below (Section 2.8). Before euthanization, blood was collected from the wing vein and processed for ELISA, as described below (Section 2.7). In anticipation of processing, the samples were stored at −20 °C. The number of chicks per trial varied depending on the number of hatched chicks, and the numbers are listed below in Table 1. As the number of chicks per hatching was limited, the in vivo trials were conducted over two time periods: in the first in vivo trials, the fertilized eggs were collected and incubated two weeks after the last booster; in the second in vivo trials, they were collected and incubated two months after the last booster. The bacterial challenge at two weeks of age was repeated to compare the results with those from the chicks incubated at two weeks after the final booster, a design choice that accounts for possible differences in antibody titers between the two time periods. Husbandry, euthanasia methods, experimental procedures and biosafety precautions were approved by the Ethical Committee (EC2023_015) of the Faculty of Veterinary Medicine, Ghent University, Belgium.

### 2.4. Bacterial Strains, Culture Conditions and Isolation

*C. jejuni* KC40 was used as challenge strain in the threshold trials. This strain was isolated at the Flanders Research Institute for Agriculture, Fisheries and Food (ILVO, Melle, Belgium) and has been used in preceding research and challenge trials, where it proved to be a high colonizer of broiler ceca [22,23,24,25,26]. The bacterin vaccine and corresponding coating of the whole-cell ELISA plates consisted of 13 *Campylobacter* strains, 11 *C. jejuni* strains and 2 *C. coli* strains, as listed in Appendix A. All strains were isolated from chicken samples [25] and provided by Sciensano (Brussels, Belgium). For the selection of these strains, genetic heterogeneity, prevalence ratio in broilers and role in human campylobacteriosis cases were considered [25]. These strains were stored at −80 °C.

*Campylobacter* bacteria were cultured on modified charcoal cefoperazone deoxycholate agar (mCCDA, CM0739; Oxoid Ltd., Basingstoke, Hampshire, UK) followed by microaerobic incubation (5% O_2_, 5% CO_2_, 5% H_2_, 85% N_2_) at 42 °C for 24 h. Next, bacteria were grown in Nutrient Broth No. 2 (NB2, CM0067; Oxoid Ltd., Basingstoke, Hampshire, UK) enriched with Modified Preston *Campylobacter*-selective supplement (SR0204E; Oxoid Ltd., Basingstoke, Hampshire, UK) and *Campylobacter*-specific growth supplement (SR0232E; Oxoid Ltd., Basingstoke, Hampshire, UK) at 37 °C for 16 h under microaerobic conditions. 

Before and after inoculation, the concentration of *Campylobacter* bacteria in the bacterial suspensions was determined by titration. Tenfold dilutions in Hank’s Balanced Salt Solution (HBSS; GIBCO-BRL, Invitrogen, Carlsbad, CA, USA) were plated on mCCDA supplemented with charcoal cefoperazone deoxycholate agar (CCDA) selective supplement (SR0155E; Oxoid Ltd., Basingstoke, Hampshire, UK) and *Campylobacter*-specific growth supplement (SR0232E; Oxoid Ltd., Basingstoke, Hampshire, UK). The plates were then subjected to microaerobic incubation at 37 °C for 48 h. 

Via spectrophotometry (Genesys 10S UV-VIS, Thermo Scientific, Waltham, MA, USA) at 600 nm, the bacterial concentration was estimated in a suspension. For the preparation of the challenge-strain inoculum, the enriched broth containing *C. jejuni* KC40 was diluted in non-enriched NB2 to the desired concentration. 

To isolate *Campylobacter* from the cecal content of breeders and broilers, samples were diluted 1:9 (*w*/*v*) in enriched NB2 and incubated at 42 °C for 24 h under microaerobic conditions. Subsequently, these samples were plated on mCCDA agar plates supplemented with CCDA selective supplement and *Campylobacter*-specific growth supplement under the same culturing conditions. After culture, *Campylobacter* colonies were identified with matrix-assisted laser desorption ionization time-of-flight mass spectrometry (MALDI-TOF MS; Autoflex Speed LRF, Bruker, Billerica, MA, USA) [27] using the direct-transfer method and using α-cyano-4-hydroxycinnamic acid (HCCA) as a matrix, according to the manufacturer’s guidelines. The spectra were obtained and analyzed with MBT Compass software version 4.1 (Bruker Daltonik, Billerica, MA, USA) and compared with a database of 6120 mean spectra projections. Identifications with a score value > 2.000 were considered reliable to the species level. The *Campylobacter* colonies were purified by plating them again on blood agar plates under the same culturing conditions. In anticipation of processing, the samples were stored at −80 °C.

### 2.5. Characterization of Campylobacter Isolates

*Campylobacter* isolates were identified and further characterized. The species was determined via MALDI-TOF MS (Autoflex Speed LRF, Bruker, Billerica, MA, USA) [27], as described above (Section 2.4). Hereafter, clonal complexes (CC) and sequence types (ST) of the strains were determined via multilocus sequence typing (MLST). Via the DNeasy Blood and Tissue kit (Qiagen, Venlo, Hilden, Germany), DNA was extracted from the isolates, and the DNA yield and quality were examined spectrophotometrically (NanoDrop, Thermo Scientific, Waltham, MA, USA). Finally, the samples were diluted to 25 ng/µL. Primers, PCR conditions, sequencing and MLST scheme were used as described by Jolley et al. [28]. The isolated strains were analyzed via MLST to determine whether the strains colonizing broiler flocks were homologous or heterologous strains compared to those colonizing the respective breeder flocks. 

### 2.6. Bacterin and Subunit Vaccine Preparation

For the preparation of the bacterin and subunit vaccine/ELISA, the protocol was used the same as that in [25], with minor adaptions. To summarize, to prepare the bacterin, the 13 *Campylobacter* strains (Appendix A) were cultured and enumerated separately in enriched NB2, as discussed above. Next, to kill the bacteria, suspensions were incubated overnight with 36% formaldehyde (0.005 mL/mL) (Sigma-Aldrich, St. Louis, MO, USA) at 37 °C. Bacterial suspensions were subsequently centrifuged for 30 min at 5718 relative centrifugal force (rcf) at room temperature. The pellets were resuspended in 5 mL of phosphate-buffered saline (PBS) with the same formaldehyde concentration (36% formaldehyde/L) mentioned above and incubated overnight at 37 °C. The next day, the successful killing of the bacteria was checked by culturing the suspensions on sheep blood agar plates under microaerobic conditions at 42 °C for 24 h. Finally, the bacterial cultures were stored at 4 °C for further use. 

Six immunodominant antigens (AtpA, CheV, Ef-Tu, GroEL, LivJ and Tig) were integrated into the subunit vaccine (Appendix A), based on preceding studies [22,24,25]. The inclusion of the recombinant proteins in the subunit vaccine was based on the reactivity of IgY from *C. jejuni* KC40 immunized layer hens [24], the association of these proteins with the bacterial cell membrane [24,29,30,31] and positive results in previous vaccination studies [22,25].

To generate the antigens sourced from the *C. jejuni* reference strain KC40, the *Escherichia coli* Expression System using Gateway^®^ Technology (Invitrogen, Carlsbad, CA, USA) was used, following the outlined methodology of Vandeputte et al. [25] with minor adaptions. Changes included the addition of 10% glycerol (Sigma-Aldrich, St. Louis, MO, USA) to the elution buffer and a subsequent dialysis step in phosphate-buffered saline (PBS) and 10% glycerol (Sigma-Aldrich, St. Louis, MO, USA) overnight after elution. These adaptations were introduced to counter protein precipitation and conserve the biological activity of the proteins.

To make the bacterin vaccine, 7 log_10_ cfu of each *Campylobacter* strain was mixed. Thus, a bacterial suspension consisting of 8.1 log_10_ cfu inactivated *Campylobacter* was obtained [22,25]. For the subunit vaccine, 75 µg of protein (12.5 µg of each recombinant antigen as listed in Appendix A) was used per dose [22,25]. The negative controls were vaccinated with a sham vaccine containing 150 µL of pure HBSS. HBSS was added to the antigen until a volume of 150 µL was reached, and the (sham) vaccine was combined with Montanide ISA 71 VG adjuvant (SEPPIC, Paris, France) at a dosage of 500 µL per chick at a 3:7 ratio.

### 2.7. Determination of IgY Titers in Egg Yolk and Serum

To determine the IgY titers in egg yolks and serum, the same protocol used by Hermans et al. [24] and Vandeputte et al. [25] was used, with some small adaptations. Prior to ELISA, the egg yolks were diluted 1/5 (*v*/*v*) in PBS and mixed. Samples were incubated overnight at 4 °C; eventually, the watery-soluble supernatant was collected for processing. After the collected blood was allowed to clot, the blood was centrifuged (1425 rcf, 15 min). Finally, the serum was collected for IgY quantification. In anticipation of processing, the samples were stored at −20 °C.

To quantify IgY titers against *Campylobacter*, both a whole-cell and a recombinant-protein ELISA were used. Overnight, 96-well flat-bottom plates (Nunc MaxiSorp, Nalge Nunc Int., Rochester, NY, USA) were coated (4 °C) with 13 different *Campylobacter* whole-cell strains (for the field screening of broiler breeders and broilers), the whole-cell *Campylobacter* KC40 strain (bacterin-vaccinated chicks’ serum collected during the threshold trial) or a mixture of 6 subunit antigens (for the subunit breeder group and subunit-vaccinated chicks). Whole-cell plates were coated with 10^6^ cfu/well, and subunit plates were coated with a mixture of 6 µg/well recombinant proteins. Both coatings were diluted in 50 µL coating buffer. After coating, plates were washed (3× washing buffer: 0.1% Tween-20 in PBS) and the wells were blocked (90 min, 37 °C) with 100 µL blocking buffer (5% skim milk powder in washing buffer). A ½ dilution series of the samples was made in blocking buffer, and 100 µL was incubated in duplo for 60 min at room temperature. The washing procedure was performed as described earlier, and the samples were incubated with 100 µL 1/10,000 horseradish peroxidase (HRP)-labelled anti-chicken IgY (Sigma-Aldrich, St. Louis, MO, USA) in washing buffer for 90 min at room temperature. Chromogenic substrate 3,3′,5,5′-tetramethyl benzidine (TMB) (Sigma-Aldrich, St. Louis, MO, USA) was used, and the plates were incubated with 50 µL at room temperature in the dark. After 10 min, 50 µL 0.5 M H_2_SO_4_ was added as stopping solution to each well and the absorbance was measured at 450 nm (OD450) using an automated spectrophotometer (Pharmacia LKB Ultrospec III, Gemini BV, Apeldoorn, The Netherlands). 

The cut-off in the dilution series, which was used to determine the IgY titer of a sample, was analyzed using the R package “changepoint” [22,32,33]. The changepoint in the dilution series of the negative control acted as cut-off value. The limit of detection was set to the dilution of the negative control at which this changepoint occurred. The IgY titer of the sample was determined based on the last dilution with an OD450 value greater than the cut-off value. When the dilution at which the changepoint took place was lower than the limit of detection, the sample was considered negative. To ensure reliability and repeatability between the different ELISA assays, the same positive-control dilution series was used, depending on the matrix (serum, yolk), the antigen (bacterin, subunit, KC40) and the age of the birds (broiler chicks, breeders).

### 2.8. Cecal Campylobacter Enumeration by qPCR

DNA was extracted using the QIAamp^®^ Fast DNA Stool Mini Kit (Qiagen, Venlo, Hilden, Germany) according to the manufacturer’s instructions, with a small adaptation. To elute the DNA, 100 µL ATE buffer was used instead of 200 µL. To test for the presence of thermophilic *Campylobacter* spp. and quantify these bacteria in the ceca, qPCR (Model CFX 96, Biorad, Hercules, CA, USA) was used, as described by Lund et al. [34] and Botteldoorn et al. [35] and applied by Vandeputte et al. [25]. The quality and the concentration of the DNA were examined spectrophotometrically (NanoDrop, Thermo Scientific, Waltham, MA, USA). Samples underwent triplicate processing, and the mean of these measurements was computed. To ensure reliability and repeatability between the different qPCR assays, the same positive-control dilution series was used.

### 2.9. Statistical Analysis

Figure 1 was made in BioRender (www.biorender.com, accessed on 28 March 2024), and the remaining figures were made in GraphPad Prism 9. In the field trial, GraphPad Prism 9 served as the analytical tool. Significance testing aimed to ascertain whether the antibody titers of breeder flocks in the MAB− and MAB+ groups differed significantly (*p* < 0.05) from each other. This assessment utilized the OD450 values obtained from ELISA at dilution 1:400 and involved a comprehensive comparison with the OD450 values of breeder flocks in the other group at the same dilution. As not all flocks had parametric OD450 values, Kruskal−Wallis was used as a test to determine significance.

The *Campylobacter* prevalence data from the monitored broiler flocks in the field study were analyzed with R.4.2.0. A generalized (i.e., binomial) linear mixed-effects regression model was used, with *Campylobacter* presence or absence as the dependent variable and age and antibody titer of the respective breeder flocks as independent variables. A random intercept term was included for each separate farm to allow the intercept to vary across trials. The significance of the impact of the maternal antibody titer on the outcome was determined via the “summary” function.

For analysis of the in vivo challenge trials, R.4.2.0 was also used. *C. jejuni* counts were transformed to log_10_ counts. To assess how *Campylobacter* colonization probability (disease prevalence) varies with treatment (i.e., control versus subunit- and bacterin-vaccinated birds), inoculation dose and week, we fitted Bayesian generalized linear models (R function ‘bayesglm’, binomial error distribution) using *Campylobacter* infection presence or absence as the dependent variable. First, analyses were run for each week and inoculation dose separately. Secondly, an overall analysis was conducted specifying treatment, inoculation dose, week and all two-way interactions between them as independent variables. As AabTiter differed between trials, this factor was included in the overall analysis as an additional covariate. Analysis of CampyTiterCounts followed a similar framework, using a Gaussian error distribution where possible (R function ‘lm’) or a zero-inflated model where needed (R function glmmTMB). Model fit and assumptions were checked using Shapiro−Wilk W values of regression residuals (i.e., W > 0.95 for a Gaussian distribution) and using glmmTMB’s built-in significance test for zero-inflation. Model selection followed a stepwise backward-selection procedure, eliminating non-significant interactions and terms until a minimal model was obtained at *p* < 0.05. Differences between the control and vaccine treatments at different inoculation doses were obtained using the ‘emmeans’ R function, using the Tukey method to adjust *p* values for multiple pairwise comparisons.

## 3. Results

### 3.1. Field Study: Trend of Decreased Susceptibility to Campylobacter Colonization in Broilers Originating from Breeders with High Antibody Levels

All sampled breeder flocks showed anti-*Campylobacter* IgY antibodies in their yolks, with pooled titers ranging from 1:200 to 1:3200 (Figure 2). No correlation was observed between the age of the flocks and the antibody titers (Spearman, *p* > 0.05). From the sampled breeder flocks, six were selected for further monitoring. The selection was based primarily on the antibody titers but also involved logistical factors (e.g., the supply of Flemish broiler farms) and the willingness of the farmer to cooperate. Four breeder flocks (D, E, P and U) with low titers were selected. The titers of these flocks were 1:680 ± 72, 1120 ± 290, 1140 ± 197 and 1:200 ± 62, respectively. Additionally, two flocks (J and W) with high antibody levels were selected, with 1:3560 ± 816 and 1:4480 ± 610 as their respective titers. The titers of the selected breeder flocks with low antibodies (MAB−) differed (*p* < 0.05) from those of the selected flocks with high antibodies (MAB+). Flock J differed significantly from flock D (*p* < 0.0001), flock E (*p* < 0.0001), flock P (*p* = 0.003) and flock U (*p* < 0.0001). Flock W also differed significantly from flock D (*p* < 0.0001), flock E (*p* < 0.001), flock P (*p* = 0.049) and flock U (*p* < 0.0001). Four weeks after the initial sampling, flocks D, E, J and W were retested to ensure that the titers in the breeder flocks’ yolks remained stable. This finding was confirmed through pooled ELISA analysis of yolk samples collected from 10 eggs per stable per flock, as discussed above (Section 2.7). Flocks P and U were not retested, as the sampling of their progeny occurred within a relatively short timeframe (<4 weeks). All selected breeder flocks tested positive for *Campylobacter* after culture (Table 2). For breeder flock E, however, no results could be obtained, as no permission was granted to take cecal samples from the farm.

Already in the second week post-hatch, one MAB− broiler flock tested positive for *Campylobacter*. Among the MAB+ broiler flocks, two were positive at 3 weeks post-hatch. At 4 weeks post hatch, the endpoint of the trial, 40% of the monitored MAB− broiler flocks and 20% of the MAB+ broiler flocks had been colonized by *Campylobacter*. This pattern of colonization resulted in an overall trend in which there were more positive MAB+ broiler flocks than MAB− broiler flocks during the first 4 weeks of life (Figure 3). The overall *Campylobacter* prevalence did not differ significantly (*p* = 0.309) between MAB− and MAB+ broiler flocks. No difference was observed in *Campylobacter* load among the different positive broiler flocks. The sampling of cecal content was limited to 4 weeks of age, as almost all broiler flocks were negative for anti-*Campylobacter* maternal antibodies at week 1. Even if some undetectable levels were present, the effect of these antibodies would have worn off by 4 weeks of age.

### 3.2. Field Study: C. coli Dominant in Breeders, an Equal Prevalence of C. jejuni and C. coli in Broilers

After they had been identified to the species level via MALDI-TOF, the *Campylobacter* isolates were further characterized via MLST. *C. coli* was found in all breeder flocks (7/7). *C. jejuni* was encountered only in the breeder flock that had been selected for immunization (1/7). *C. jejuni* and *C. coli* were equally prevalent (both 3/6) in positive broiler flocks. In this study, no broiler flocks were colonized with the same ST as their respective breeder flock. The most frequently encountered clonal complex (CC) was CC828 (*C. coli*), which had colonized 6 out of 7 breeder flocks and 2 out of 6 broiler flocks. Although the same clonal complexes were found in different barns of parent flocks U and W, the same sequence type was not found across all barns. Two of the broiler flocks had been colonized by the same CC as their respective parent flock samples, with one being MAB+ and the other being MAB−. However, the STs were not the same, indicating colonization by a different strain (Table 2). The *Campylobacter* isolates derived from breeder flock J and P were not further typed, as none of their offspring was positive for *Campylobacter*. 

### 3.3. Field Study: Low Maternal Antibody Titers in Broilers from Breeders with Varying Antibody Levels

Throughout the early stages of life, pooled serum samples from the progeny flocks were subjected to weekly analysis. Almost all pooled serum samples from the different broiler flocks proved to be negative for anti-*Campylobacter* antibodies at the different timepoints, with the exceptions of flock J1 at week 1 (1:400) and flock W4 (1:200) at week 2. It is noteworthy that these two broiler flocks both originated from MAB+ breeder flocks. As all samples were negative at three weeks of age, serum was not further examined after this timepoint.

### 3.4. Vaccination Study: Immunization of Seropositive Broiler Breeders Results in Temporarily Higher Anti-Campylobacter Titers in Broiler Breeders

As already mentioned above, the seropositive breeder flock selected for vaccination (flock Delta, average yolk titer 1:2000 ± 400, CV 63%) proved to be positive for both *C. jejuni* and *C. coli*. The *C. jejuni* CCs included 677 (ST-764) and 353 (ST-7571), and the *C. coli* CC included 828 (ST-860).

Immunization of *Campylobacter*-seropositive breeders led to an increase in the antibody titers in yolk in both subunit- and bacterin-vaccinated breeders. A peak of antibody titers was reached in both subunit- and bacterin-vaccinated breeders after the second booster, and titers started to decline two months after the third booster. After this point, the titers seemed to stabilize for the subunit vaccine but continued to decrease for the bacterin vaccine until the final sampling point (4 months after the final booster). At all timepoints after vaccination, titers in the yolks of both groups were higher than in the yolks from the seropositive breeders vaccinated with the sham vaccine. At peak titers, 64 times more antibodies were measured in the yolks from the bacterin-vaccinated breeders than in the yolks from the negative control. For the subunit-vaccinated animals, 640 times more antibodies were measured in the yolks (Figure 4, Appendix A). It is imperative to note that two breeders of the bacterin-vaccinated group died 2 and 3 months after the last booster. Notably, neither case appeared to be linked to the immunization process; instead, both were attributed to fatty liver hemorrhagic syndrome. Moreover, no adverse effects, local or systemic, were observed in the months subsequent to immunization.

### 3.5. Vaccination Study: Protective Efficacy of Bacterin and Subunit Derived Maternal Antibodies on Cecal Colonization in Broilers Using a Threshold Model

In order to evaluate the susceptibility of broiler chicks to *Campylobacter* colonization during the early stages of life and to investigate the influence of maternal antibodies, a threshold model was employed following the immunization of seropositive broiler breeders. After the first incubation period (two weeks after the last booster),when the animals were challenged at one week of age with a low or a high *Campylobacter* inoculation dose, no differences were observed with regard to the presence of *Campylobacter*-positive animals in the different experimental groups (subunit, bacterin and control) (*p* > 0.05) (Figure 5). It is noteworthy that the majority of animals (>60%) were colonized and that the mean number of cecal genomic equivalents (GE) of the control group was lower than those of the bacterin (*p* = 0.015) and subunit (*p* = 0.038) groups following challenge with a low dose (Figure 6). Over the two incubation periods of the fertilized eggs (two weeks and two months after the last booster), the antibody titers decreased considerably in immunized breeders and their offspring. When looking at the results separately, only a reduction (*p* = 0.049) in *Campylobacter* prevalence could be noted in bacterin-vaccinated chicks compared to control chicks when fertilized eggs were incubated two weeks after the last booster and challenged with a high dose at week 2 (Figure 5). No other statistically significant differences were observed (*p* > 0.05). However, when we take into account the results from week 2 overall, including both the chicks inoculated at two weeks and those inoculated at two months after the last booster, and account for the differences in antibody titers, higher levels of antibody titers are significantly linked with a lower prevalence of *Campylobacter*. This is true both for bacterin versus control (*p* = 0.008) and subunit versus control (*p* = 0.003). On analysis of level of colonization, no such differences were observed (*p* > 0.05). In chicks challenged at three weeks, no differences in prevalence could be observed between the different groups (*p* > 0.05) (Figure 7). A higher cecal GE number was observed in the subunit broilers in comparison with the control (*p* = 0.031) and bacterin (*p* = 0.005) broilers when the animals were challenged with a high dose (Figure 8).

In the first in vivo trials, eggs were incubated two weeks after the final booster. The control chicks were positive for anti-KC40 antibodies (1:600) in the first week and for antibodies against both the KC40 (<1:400) and recombinant antigens (<1:1600) in the second week. In the first week, antibodies against the recombinant antigens could not be detected. The pooled anti-KC40 antibody titer in bacterin-vaccinated chicks proved to be twice as high in week 1 (1:16,000) in comparison with week 2 (1:8000). The pooled titers against the recombinant antigens in subunit-animals were 1:48,000 in week 1 and 1:19,200 in week 2 (Table 3).

For the second in vivo trials, eggs were incubated two months after the final booster. Antibodies against KC40 were observed only at week 2 in the control group. A sharp decrease in levels of both anti-KC40 and anti-subunit antibodies could be noted in the bacterin-vaccinated and subunit-vaccinated chicks, respectively, at 2 weeks of age in comparison with the chicks from the first in vivo trials. For the bacterin-vaccinated chicks, this decrease in the pooled titer was five-fold (1:1600), and for the subunit-vaccinated chicks, three times fewer (1:6400) antibodies were detected. At 3 weeks of age, antibodies were still found in bacterin- and subunit-vaccinated chicks. These pooled titers were quantified at 1:600 and 1:2400 respectively (Table 3). 

## 4. Discussion

In this study, the potential role of maternal antibodies in the protection of chicks against *Campylobacter* in the first weeks of life was investigated. In a preceding study [22], the role of maternal antibodies was demonstrated by immunizing seronegative SPF breeders with a bacterin vaccine and a subunit vaccine and subsequently challenging their offspring with different doses of *Campylobacter* in a controlled environment. In that study, a protective effect of the maternal antibodies was observed during the first 1 (bacterin) and 2 (subunit) weeks of life. Our results confirm the (very) high prevalence of *Campylobacter* in breeder flocks in the field [36], which was reflected in a high seroprevalence. Thus questions remained: whether antibodies elicited in broiler breeders by natural colonization and passed on to their progeny confer protection against in-field colonization in broilers; and whether immunization of naturally colonized broiler breeders can increase the protection in their offspring against a controlled challenge with *Campylobacter* due to elevated maternal antibody titers. 

Previous studies [19,20,22] were able to demonstrate the protective effect of maternal antibodies in broilers against *Campylobacter* colonization under experimental conditions. However, so far, no such studies have been conducted under commercial field conditions. To explore to what extent naturally elicited maternal antibodies might confer protection to broilers in the field, a field study was conducted in which broiler flocks from different breeder flocks with high or low anti-*Campylobacter* IgY levels were monitored. All sampled breeder flocks (*n* = 25) tested positive for *Campylobacter* antibodies, indicating widespread exposure to *Campylobacter* in breeder flocks. In all breeder flocks that were subsequently selected for further follow-up of their progeny and for which permission was granted to take faecal samples for further analysis, *Campylobacter* could be isolated. The high *Campylobacter* seroprevalence and colonization rates in broiler breeders observed in the current study are in line with the results of previous samplings [22,36]. Significant differences in titers made it possible to divide the breeder flocks into a group with high serum anti-*Campylobacter* IgY titers and a group with low titers. The presence of low antibody titers in combination with ongoing colonization could possibly be attributed to colonization by a less immunogenic strain or to an older colonization. In another study [20], a substantial anti-*Campylobacter* serum-antibody titer was observed one week after experimental oral inoculation of Ross breeders and layers. However, these titers seemed to have dropped a few weeks after initial colonization, although they increased upon rechallenge with *Campylobacter*.

According to the results of monitoring the offspring of MAB+ and MAB− breeder flocks, broiler flocks with lower anti-*Campylobacter* IgY titers tended to be more susceptible for *Campylobacter*, as a higher percentage of positive flocks was recorded early in the study (at 2 weeks of age) and at the endpoint of the study (at 4 weeks of age). These results were not statistically significant, possibly due to the limited sample size. The serological investigations demonstrated that almost all sampled broilers, including both those originating from both MAB− and MAB+ breeder flocks, were negative for anti-*Campylobacter* antibodies at the first sample point (one week of age). Antibodies might still have been present, but if so, they were undetectable by our ELISA method used. The changepoint observed in the dilution series of the negative control on ELISAs, occurring either at 1:200 or 1:400, suggests that antibody titers below this threshold were undetectable. In other words, the detection limit ranged between 1:200 and 1:400 [22,32,33]. Other studies investigating anti-*Campylobacter* antibodies in broilers’ serum after vaccination reported the same starting dilutions on ELISA (1:200), indicating detection limits similar to that observed in our study [37,38,39]. Although they were not significant, the results of the field study indicate possible early protection by maternal antibodies against *Campylobacter* colonization. However, as maternal antibody titers decreased rapidly and as *Campylobacter*-positive MAB+ flocks were observed, the protection thus obtained is most likely only partial. Numerous other factors likely contribute to the early onset of colonization, including biosecurity measures [16,40,41], flock size [42,43], seasonality and weather conditions [42,44], the presence of other livestock [45] and the current intestinal microflora [18,46,47]. 

Not many studies have investigated the prevalence and species prevalence of *Campylobacter* in broiler breeders. In our study, all sampled breeder field flocks tested positive for *C. coli*. Co-colonization with two *C. jejuni* strains was observed only in the breeder flock selected for immunization. In previous studies conducted in Belgium [36] and Spain [48], a 100% prevalence of *Campylobacter* in breeder flocks was also observed. However, in these studies, *C. jejuni* was observed as the predominant species, followed by *C. coli*. In contrast, our current findings indicate a noteworthy shift in Belgium, with *C. coli* currently assuming the role of the dominant species in breeder populations.

In the monitored progeny broiler flocks, *C. jejuni* and *C. coli* were equally prevalent. Just as in breeders, more *C. coli* strains were encountered than expected, as *C. jejuni* is generally considered the dominant *Campylobacter* species in broilers, both in Belgium [36,40] and in Europe [41,42,43]. Lately, there have also been reports of *C. coli* being more frequently recovered from poultry meat in Europe [44,45,46]. However, as pointed out by Andritsos et al. [47], multiple factors should be taken into account when interpreting these data, such as seasonality of the sampling, broiler age, sampling method and isolation procedure. In the study by Babacan et al. [48], it was found that birds slaughtered earlier were more frequently colonized by *C. jejuni* compared with birds slaughtered at a later timepoint, which were more frequently colonized by *C. coli.* El-Shibiny et al. [49] observed that a *C. jejuni* strain and a *C. coli* strain could both colonize and compete with each other equally until the birds were 35 days old. After this timepoint, *C. jejuni* numbers decreased, leaving the *C. coli* strain as the dominant isolate. It should be noted that these results could be strain-specific and that they do not explain the high prevalence of *C. coli* in broilers in this study, as all birds were sampled at a young age. However, the results could potentially explain the high prevalence of *C. coli* in the breeders. The shift from *C. jejuni* to *C. coli* holds significance, as it is hypothesized that *C. coli* harbors fewer virulence genes affecting humans [50]. Nonetheless, the actual impact of an elevated ratio on reducing cases of human campylobacteriosis remains a subject requiring further investigation. It would also be interesting to continue monitoring whether *C. coli* has indeed expanded its foothold in Belgium and, by extension, in Europe. If so, the reasons behind this change would be interesting to investigate. An interesting factor to explore is the direct effect of climate change and changing weather patterns. As the season-dependent patterns of *C. jejuni* and *C. coli* appear to differ from each other [42,51,52], this could be a contributing factor. 

The analysis via MLST of the colonizing strains in breeders and their offspring was intended to further explore the correlation between maternal antibodies and their potential protective effect against *Campylobacter* colonization. For example, early colonization of maternal-antibody-negative (MAB−) broiler flocks with homologous strains and its absence in maternal-antibody-positive (MAB+) broiler flocks would support the hypothesis of a potential protective effect of maternal antibodies, whereas the opposite finding would challenge this hypothesis. However, as breeder flocks and their progeny were colonized by different strains, no further conclusions could be drawn regarding the role of maternal antibodies in the protection of broiler flocks against *Campylobacter* colonization. Despite the assumptions of Cox et al. [13] about vertical transmission, no identical *Campylobacter* STs were found in breeders and their offspring in the field study. Also, in the offspring of our (sham-)immunized breeder flock from the field, which were colonized by three different *Campylobacter* strains, no *Campylobacter* was found before the experimental challenge using culture techniques. The claim could be made that these chicks had maternal antibodies and the *Campylobacter* population was suppressed by these, which made the *Campylobacter* undetectable by conventional culture methods, but even at 3 weeks of age, when maternal antibodies had decreased significantly in the control group, no *Campylobacter* was found. It is possible that *Campylobacter* could survive on the eggshells (and maybe in the outer membranes), but the assumption that this could lead to the full-blown colonization of newly hatched chicks is too far-fetched. Based on the MLST analysis, six out of seven breeder flocks and two out of three broiler flocks were colonized by *C. coli* CC828 complex, which also colonized the breeder flock selected for the immunization trial. This was no surprise, as CC828 has been recognized worldwide as the predominant clonal complex, originating from different clinical and environmental sources [53,54,55,56,57,58]. This CC has successfully established itself in an agricultural niche and subsequently causes most human campylobacteriosis cases attributed to *C. coli.* In other Belgian studies on *C. coli* in broiler carcasses and diarrheal patients [59,60], most (>80%) of the isolates belonged to CC828. The most frequently observed STs were ST-854, ST-825, ST-860, ST-5163, ST-829, ST-832 and ST-872. In our study, ST-832 was present in two broiler flocks and ST-860 was present in the breeder flock selected for immunization, but the other STs were not encountered. ST-828 was the predominant ST in our study, with three identified breeder isolates. Regarding *C. jejuni,* other Belgian studies [59,61,62] have mentioned CC-21 as one of the most common STs. Clonal Complexes, both in human and poultry samples, although the *C. jejuni* population is much more diverse. Within this CC, ST-50 and ST-21 are considered the most dominant STs. Although the number of *C. jejuni* strains in our study was limited, CC-21 was encountered in two broiler flocks, with one strain belonging to ST-50. CC-677 (ST-764) and 353 (ST-7571), isolated from the breeder flock selected for the immunization trial, have not been described in previous Belgian studies. To conclude, most of our findings regarding MLST types in *C. jejuni* and *C. coli* are in line with the findings of previous studies.

In the immunization trial, the seropositive breeder flock selected for immunization tested positive for 3 strains of *Campylobacter*, namely *C. jejuni* CC-677 (ST-764) and CC-353 (ST-7571) and *C. coli* CC-828 (ST-860), which might explain the strong serological reaction on ELISA. One strain even belonged to the same CC as our *C. jejuni* challenge strain (KC40; CC-677 ST-794). Next, this seropositive breeder flock was immunized using a bacterin vaccine and a subunit vaccine. The selection of *C. jejuni* and *C. coli* strains incorporated into the bacterin vaccine was initially based on their prevalence in poultry and in human campylobacteriosis cases [25]. The bacterin contains strains belonging to the same clonal complexes encountered in the current study, except for CC-353, supporting their relevance. Immunization with both the bacterin vaccine and the subunit vaccine resulted in increased anti-*Campylobacter* titers in the yolks compared to the yolks from sham-immunized seropositive breeders. The maximum titers in both bacterin-vaccinated and subunit-vaccinated breeders were reached after the second booster, which suggests that the third booster might not be necessary. However, these titers proved unstable, as after the second and third booster, they started to decline. Despite this decline, they were still higher than the titers of the sham-immunized breeders. 

Next, to assess the protective efficacy of the increased maternal antibodies against *Campylobacter* colonization, a threshold challenge model was conducted. No trends or differences in prevalence were observed between the different groups following challenge at weeks 1 and 3. It is possible that the challenge dose was too high in week 1, even for the lower challenge dose, as the lower challenge dose was slightly higher than that in the preceding study by Haems et al. [22] (3.0 log_10_ versus 2.5 log_10_), in which a protective effect of maternal antibodies elicited by immunization was observed. Additionally, more than 64% of each group was colonized, which also points towards a strong challenge. In week 3, maternal antibodies had possibly decreased too much to make a difference, although subunit-vaccinated animals had a higher *Campylobacter* load in their caeca than did the two other groups following challenge with a high dose. If we account for the differences in antibody levels and examine the overall results from week 2, a significant link between decreased prevalence in bacterin-vaccinated chicks and subunit-vaccinated chicks and their respective antibody levels can be observed. This finding confirms that in week 2, both bacterin- and subunit-elicited antibodies played a role in the protection of chicks against *Campylobacter* challenge. However, in a previous study in which SPF breeders were immunized [22], the protective efficacy of the maternal antibodies induced by the bacterin and subunit vaccines proved to be more significant [22]. This difference can be explained by the higher baseline levels of antibodies present in the breeders and their progeny, as the SPF breeders in the study of Haems et al. [22] were seronegative. The in-field breeders used in this study were colonized by a *Campylobacter* strain belonging to the same clonal complex as the challenge strain, which provided the sham-immunized breeders and their chicks with homologous humoral protection. Although it was proven in a previous study [22] that immunization of SPF breeders was able to increase the level of anti-*Campylobacter* IgY antibodies in the intestinal mucus, the enhanced maternal immunity induced by the vaccines in seropositive breeders had only a limited effect with regard to protection against *Campylobacter* colonization in week 2 in both subunit- and bacterin-vaccinated chicks. 

Finally, an important factor to consider is the way birds were challenged. It was suggested by other studies [63] that the handling of the chicks may induce stress and that the experimental challenge with a pure *Campylobacter* culture at a high dose may disrupt the native microflora, potentially facilitating easier colonization by *Campylobacter*. However, a comparative study between different treatment groups requires that birds be inoculated with a standardized challenge dose, ensuring scientific rigor in drawing conclusions regarding treatment efficacy. To address this necessity, the lowest challenge dose at which *C. jejuni* KC40 strain maximally colonizes broiler chicks from SPF breeders was chosen as the low challenge dose (unpublished data [22]). However, the possibility that the experimental conditions themselves may have favored *Campylobacter* colonization, thereby potentially obscuring the true treatment effect, cannot be excluded. Despite this possibility, significant *Campylobacter* reductions were also obtained with this strain at much higher challenge doses (8 × 10^3^–10^5^ cfu) when chicks were orally supplemented with yolk from hyperimmunized laying hens [24,25].

## 5. Conclusions

To conclude, seropositivity for *Campylobacter* in breeder flocks is widespread, indicating regular contact of these flocks with *Campylobacter*. Additionally, *C. coli* was the most commonly isolated species both in broiler and breeder flocks, indicating a potential shift from *C. jejuni* to *C. coli* as the dominant species on Belgian poultry farms. On farms, chicks with higher levels of maternal antibodies tend to be less susceptible to *Campylobacter* colonization in the first weeks of life. However, taking into account the rapid decline of titers in the serum of the chicks, maternal antibody transfer probably offers only partial protection. Using immunization, it was possible to increase anti-*Campylobacter* antibody titers in the serum of seropositive breeders and consequently also in the serum of the offspring. Indications for improved resistance against *Campylobacter* colonization in offspring from these immunized breeds are limited and inconsistent. All findings taken into account, the presumed protective effect of maternal immunity should be considered limited and should not be overestimated in the context of *Campylobacter* control in broilers.

## Figures and Tables

**Figure 1 animals-14-01291-f001:**
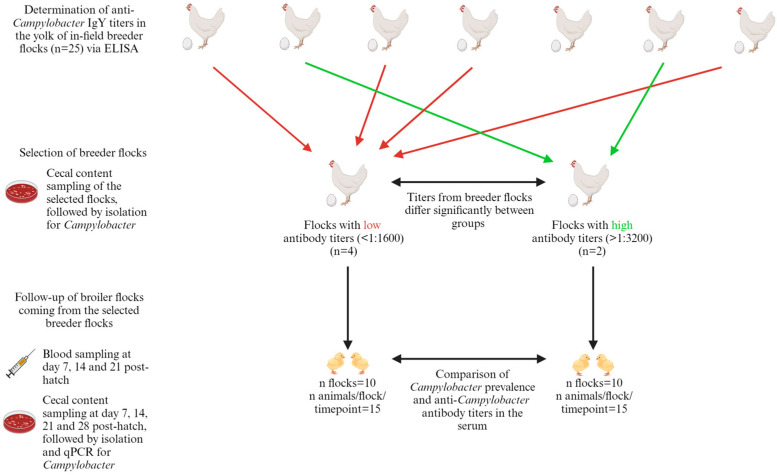
Schematic overview of the field study, where broiler flocks coming from breeder flocks with high or low anti-*Campylobacter* antibody titers in the yolk were sampled for the presence of *Campylobacter* and anti-*Campylobacter* IgY antibodies in the first 4 weeks post-hatch.

**Figure 2 animals-14-01291-f002:**
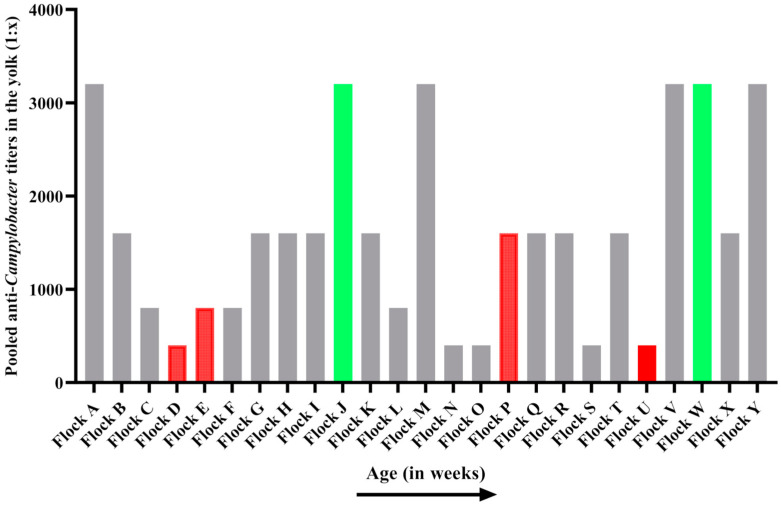
Anti-*Campylobacter* IgY titers in pooled yolks from 25 broiler breeder field flocks ranked by age. Flocks with statistically significant high titers (in green) and flocks with statistically significant low titers (in red) were selected for follow up of their progeny. Flock J differed significantly from flock D (*p* < 0.0001), flock E (*p* < 0.0001), flock P (*p* = 0.003) and flock U (*p* < 0.0001). Flock W also differed significantly from flock D (*p* < 0.0001), flock E (*p* < 0.001), flock P (*p* = 0.049) and flock U (*p* < 0.0001). Flocks that were not selected are represented in grey.

**Figure 3 animals-14-01291-f003:**
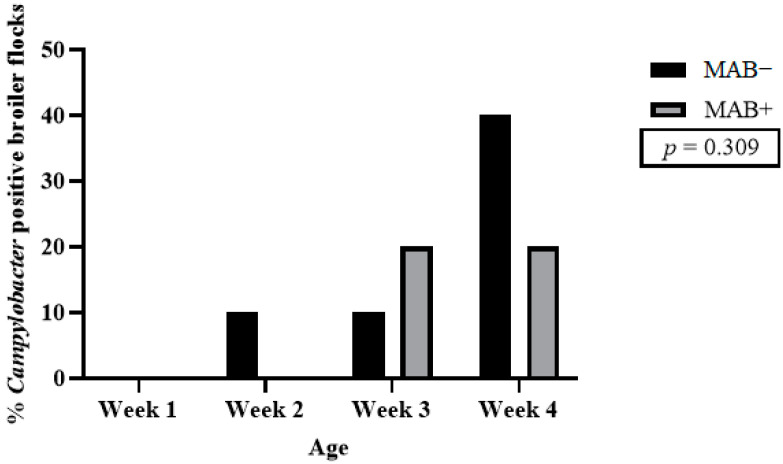
Percentage of *Campylobacter* positive broiler flocks at each sample point, as determined by isolation from pooled cecal content and confirmed by qPCR. The broiler flocks are progeny from breeders with low anti-*Campylobacter* IgY yolk antibodies (MAB−) or high anti-*Campylobacter* IgY yolk antibodies (MAB+).

**Figure 4 animals-14-01291-f004:**
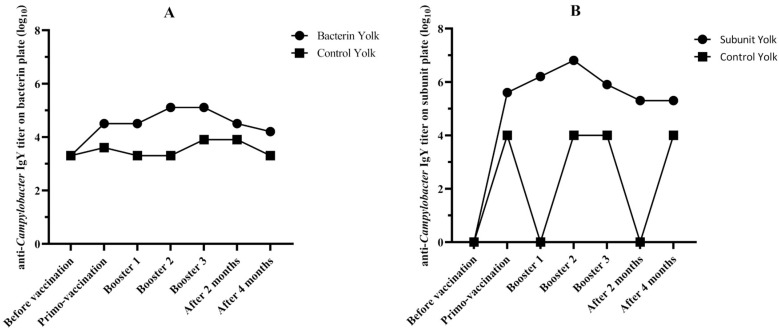
Evolution of anti-*Campylobacter* IgY titers in the yolks of broiler breeders vaccinated with the bacterin (**A**), the subunit (**B**) and the negative control (**A**,**B**) vaccine as determined by ELISA on the plates coated with the bacterin antigen (**A**) and the antigen subunit (**B**). For comparison, titers of the seropositive breeders vaccinated with the sham vaccine against the bacterin antigen and subunit antigen are included.

**Figure 5 animals-14-01291-f005:**
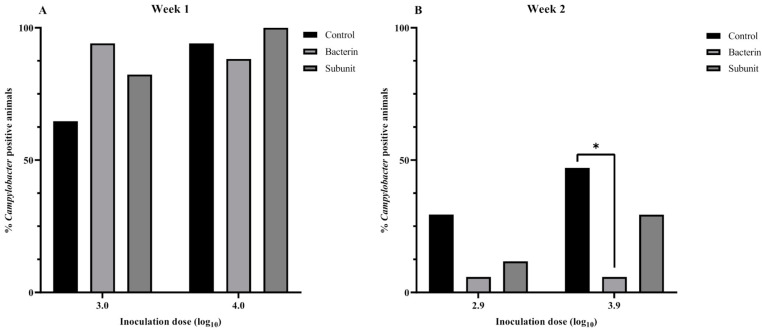
Percentage *Campylobacter*-positive broilers after experimental inoculation with *C. jejuni* strain KC40 at the ages of 1 (**A**) and 2 (**B**) weeks using a threshold model. The fertilized eggs of these chicks were incubated two weeks after the last booster. The individually housed broilers were progeny from *Campylobacter*-positive broiler breeders immunized with a bacterin or subunit vaccine. The positive controls were progeny from *Campylobacter*-seropositive broiler breeders vaccinated with the sham vaccine. The statistical significance of differences between groups are summarized as * *p* ≤ 0.05. When no symbol representing statistical significance is displayed between groups, *p* ≥ 0.05.

**Figure 6 animals-14-01291-f006:**
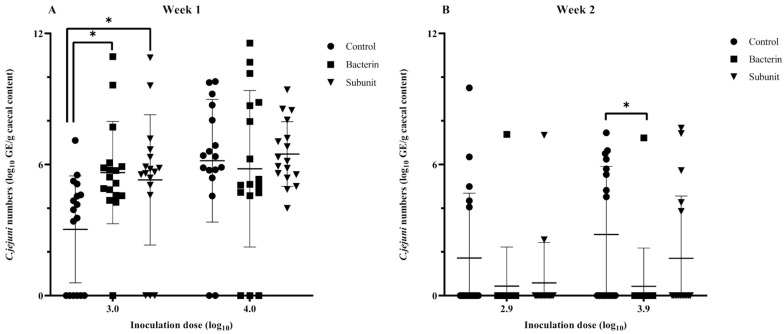
Individual and mean numbers of cecal *Campylobacter* genomic equivalents (log_10_ GE/g) in broilers inoculated with a fixed dose of *C. jejuni* strain KC40 at the ages of 1 week (**A**) and 2 weeks (**B**) using a threshold model. The fertilized eggs of these chicks were incubated two weeks after the last booster. The individually housed broilers were progeny from broiler breeders immunized with a bacterin or subunit vaccine. The positive controls were progeny from sham-immunized broiler breeders. The statistical significance of differences between groups are summarized as * *p* ≤ 0.05. When no symbol representing statistical significance is displayed between groups, *p* ≥ 0.05.

**Figure 7 animals-14-01291-f007:**
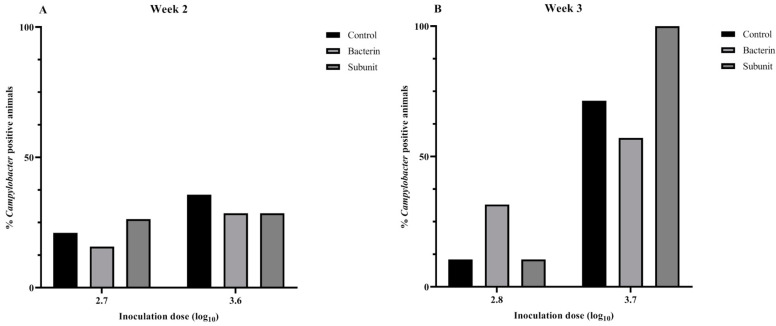
Percentage *Campylobacter*-positive broilers after experimental inoculation with *C. jejuni* strain KC40 at the ages of 2 (**A**) and 3 (**B**) weeks using a threshold model. The fertilized eggs of these chicks were incubated two months after the last booster. The individually housed broilers are progeny from broiler breeders immunized with a bacterin or subunit vaccine. The positive controls are progeny from sham-immunized broiler breeders. When no symbol representing statistical significance is displayed between groups, *p* ≥ 0.05.

**Figure 8 animals-14-01291-f008:**
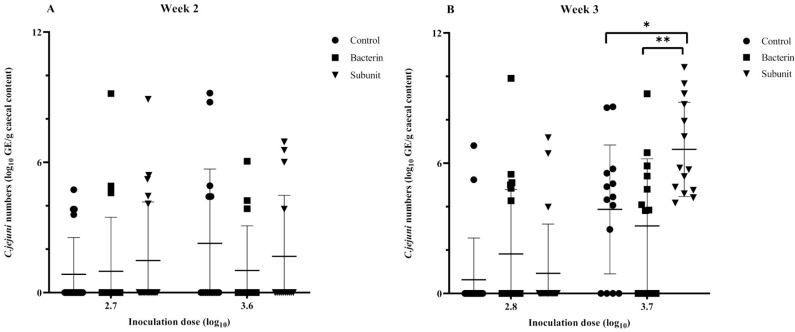
Individual and mean numbers of caecal *Campylobacter* genomic equivalents (log_10_ GE/g) in broilers inoculated with a fixed dose of *C. jejuni* strain KC40 at the age of 2 weeks (**A**) and 3 weeks (**B**) using a threshold model. The fertilized eggs of these chicks were incubated two months after the last booster. The individually housed broilers are progeny from broiler breeders immunized with a bacterin or subunit vaccine. The positive controls are progeny from sham-immunized broiler breeders. The statistical significance between groups is summarized as * *p* ≤ 0.05, ** *p* ≤ 0.01. When no statistical significance is displayed between groups, *p* ≥ 0.05.

**Table 1 animals-14-01291-t001:** Summary of the threshold trials design.

Time of Inoculation	Challenge Dose in log_10_ cfu	Sample Size per Group (*n*)
Week 1 *	3.0	Bacterin, subunit, control (*n* = 17)
	4.0	Bacterin, subunit, control (*n* = 17)
Week 2 *	2.9	Bacterin, subunit, control (*n* = 17)
	3.9	Bacterin, subunit, control (*n* = 17)
Week 2 ^‡^	2.7	Bacterin, subunit, control (*n* = 19)
	3.6	Bacterin, subunit, control (*n* = 14)
Week 3 ^‡^	2.8	Bacterin, subunit, control (*n* = 19)
	3.7	Bacterin, subunit, control (*n* = 14)

* = the fertilized eggs were incubated 2 weeks after the last booster. ‡ = the fertilized eggs were incubated 2 months after the last booster.

**Table 2 animals-14-01291-t002:** The clonal complex (CC) and sequence type (ST), as determined by MLST analysis of the *Campylobacter* strains isolated from the in field-monitored breeder and broiler flocks. These flocks included breeder flocks with high (≥1:3200) anti-*Campylobacter* IgY titers (MAB+), breeder flocks with low (≤1:1600) titers and their respective progeny flocks.

MAB−							
Breeder Flock	Species	CC	ST	Broiler Flock	Species	CC	ST
D (Pooled)	*C. coli*	UA *	6549	D1	*C. jejuni*	21	21
E	No permission granted			E1	*C. jejuni*	UA	2274
				E2	*C. coli*	UA	10,044
				E3	Negative		
				E4	Negative		
P (Stable 1)	*C. coli*	Not analyzed *		P1	Negative		
P (Stable 2)	*C. coli*	Not analyzed		P2	Negative		
				P3	Negative		
				P4	Negative		
U (Stable 1)	*C. coli*	828	UA	U1	*C. coli*	828	832
U (Stable 2)	*C. coli*	828	828				
**MAB+**							
**Breeder Flock**	**Species**	**CC**	**ST**	**Broiler Flock**	**Species**	**CC**	**ST**
J (Pooled)	*C. coli*	Not analyzed		J1	Negative		
				J2	Negative		
W (Stable 1)	*C. coli*	828	1058	W1	*C. coli*	828	832
W (Stable 2)	*C. coli*	828	828	W2	*C. jejuni*	21	50
W (Stable 3)	*C. coli*	828	828	W3	Negative		
				W4	Negative		
				W5	Negative		
				W6	Negative		
				W7	Negative		
				W8	Negative		

* UA = Unassigned; the isolate could not be assigned to a specific CC or ST. * Not analyzed; as none of the offspring broiler flocks was positive and no link could be made between the strains colonizing the breeder and the broiler flock, the strain was not further characterized.

**Table 3 animals-14-01291-t003:** Pooled serum IgY titers present in bacterin- and subunit-vaccinated chicks versus control chicks, challenged with *C. jejuni* strain KC40 in threshold trials and quantified via ELISA. Birds were sampled when they were euthanized, two days after inoculation. Bacterin-vaccinated, subunit-vaccinated, and control chicks were the offspring of bacterin-, subunit- and sham-vaccinated breeders, respectively. Control samples were analyzed both on a KC40 whole-cell coating and on a subunit coating.

Group	Inoculation Dose (log_10_ cfu)	KC40 Whole-Cell Coating		Subunit Coating	
		Bacterin	Control	Subunit	Control
Week 1 *	3.0	1:16,000	1:400	1:32,000	Negative
	4.0	1:16,000	1:800	1:64,000	Negative
Week 2 *	2.9	1:8000	Negative	1:25,600	Negative
	3.9	1:8000	Negative	1:12,800	Negative
Week 2 ^‡^	2.7	1:1600	Negative	1:6400	Negative
	3.6	1:1600	1:400	1:6400	Negative
Week 3 ^‡^	2.8	1:400	Negative	1:3200	Negative
	3.7	1:800	Negative	1:1600	Negative

* = the fertilized eggs were incubated 2 weeks after the last booster. ‡ = the fertilized eggs were incubated 2 months after the last booster.

## Data Availability

The data presented in this study are openly available in Open Science Network at DOI 10.17605/OSF.IO/GHT7F, reference number ght7f.

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
