# Peer review of "Role of Maternal Antibodies in the Protection of Broiler Chicks against Campylobacter Colonization in the First Weeks of Life"

_animals, 2024, doi:10.3390/ani14091291_

Round 1

Reviewer 1 Report

Comments and Suggestions for Authors

Campylobacter control in poultry is still a major area of interest across the industry and so this manuscript is investigating an important area.  Studies with Campylobacter are difficult as it is a challenging bacterium with a complex relationship with its host.  The data from this manuscript is a good example of that complexity.  On the whole the manuscript is good albeit a little hard to follow in places as the authors have combined multiple studies - one suggestion would be to try and make the different studies more distinct. 

I have a few specific suggestions for consideration

Line 49 - IS there a more up to date figure?
Line 62-63 - A reference here would be good.  Maybe include some papers where control methods have been reviewed
Line 67-69: Faecal contamination on the egg would be considered horizontal by some and not true vertical transmission. Vertical transmission is typically considered to be presence of the bacterium inside the egg before laying.
Line 123-124:  Why the highest titre? *maybe not this comment*
Line 127-128: Why were they screened for campylobacter?
Line 131: What was the feeding schedule?
Line 163: HOw were the chicks innoculated with Campylobacter?
Line 242-243: REferences for the "preceding studies"
Lines 349-351: Where is that data confirmed?
Line 415-419: Where the authors discuss the peaks and declines in antibodies - is this both serum and yolk? If so was the rate of decline similar in both matrices?
Line 419-421: Was this difference significant? Where is this data presented?
Line 561: Why might they not be detected? Give some references regarding ELISA sensitivity and why your test may not be sensitive enough.
Line 561: extra space between ELISA and method.
Line 568-570: revise English
Line 575-576: revise English
Line 577-579: Any reason why this may be the case and is this important?
Line 583: Extra space after "However,"
Line 594: Are the antibodies for C. jejuni cross reactive with C. coli? Could the shift in C. jejuni and C. coli populations be related to antibody production?
Line 596-598: The authors could comment here on vertical transmission.  Different strains in breeders and offspring would intimate that there is no vertical tranmission.
Line 633-657: In this section there a few things to consider. For the challenge dose - how does it compare to other studies? What is the level of campylobacter in the environment where the chick have a natural challenge.  One could also consider that artifical challenge with a pure culture of campylobacter could be stressful for the chicks (due to the innoculation) which can boost campylobacter activity and thus the expected protection from antibodies may not be seen (See 10.3382/ps/pey295 where this concept is discussed).  Another possibility could be that the antibodies for campylobacter are not secreted/expressed at high levels into the gut lumen thus there is limited opportunity for them to help reduce campylobacter colonisation?

Line 662-664: This is not really true.  Out of the comparisons in figures 5, 6, 7 and 8 there are only a handful of significant differences (some of which show less campylobacter in control birds) and the trends are not consistent.  
Figure 3 legend: Consider putting the p-value to highlight that the differences are not significant.
Figure 4: Error bars and significance values would be beneficial in this figure.  The data in this figure is a little confusing - Why do the authors think that the data for the control yolk in B is so different to the control yolk in A? Surely the IgY titres should follow the same pattern? Also, why do they propose the time points where it drops to 0 and then back to 4?
Table 3: Maybe put the row with "KC40 whole cell.... subunit coating" above the "bacterin control subunit control" to make it clearer what each column is

Comments on the Quality of English Language

There are a few areas where informal English is used or the sentence structure is not correct. I'd recommend checking through the whole document again.

Author Response

Dear Reviewer,

Thank you for your insightful suggestions. Please see the attachment.

Kind regards,

the Authors

Reviewer 2 Report

Comments and Suggestions for Authors

Line 106:  How did authors pool serum samples?

Line 131: What is the control group? Placebo-control?

Line 164: The study did use Campylobacter strain KC40. Please define a species of Campylobacter.

Author Response

(The authors gave the same response as above.)

Reviewer 3 Report

Comments and Suggestions for Authors

Line 16: Remove the full stop after "to."

Line 52: Replace "C. jejuni" with "Campylobacter (C.) jejuni."

Figure 1: Check the number of flocks with low antibody titers (n=4) or (n=3) for accuracy.

Line 166: Specify the location of the qPCR analysis referred to.

Line 168: Clarify whether the information is mentioned above or below.

Line 188: Ensure the information listed in Table 1 matches the description in the text.

Line 207: Correct the format of scientific names to italicize them throughout the text.

Line 220: Change "ng/µl" to "ng/µL."

Line 261: Adjust the format of "500µl" to "500 µL" throughout the text.

Line 402: Review and correct the sentence accordingly.

Materials and methods: Add suitable references for each experiment and include models and company names of equipment used.

Ensure consistency in using "described above" or "described below," and mention the corresponding heading numbers in parentheses.

Maintain the correct format for temperature (e.g., 37°C) and percentage (e.g., 36%).

Figure 2: Verify that the X axis accurately represents breeder flocks, not just age.

Include statistical values and associations in the text, tables, and figures.

Figure 4: Include values for the control group.

Ensure that the results align with the experiments and rewrite them accordingly for clarity and coherence.

Additional Comments:

The main question addressed by the research is the potential protective role of maternal antibodies against Campylobacter colonization in broiler chicks. The study aims to investigate whether the presence of anti-Campylobacter antibodies in broiler breeders correlates with reduced susceptibility to Campylobacter colonization in their offspring.

The topic is both original and highly relevant in the field of animal health and food safety. Effective measures to prevent Campylobacter colonization in broiler flocks are currently lacking. This study addresses a specific gap in the field by focusing on the potential role of maternal antibodies in protecting broiler chicks against Campylobacter colonization.

This research adds significant value to the subject area compared to other published material by providing novel findings on the role of maternal antibodies in protecting broiler chicks against Campylobacter colonization.

It would be beneficial for the authors to further elaborate on the standardization of sampling protocols to ensure consistency and minimize variability between samples. This could include specifying the exact procedures for sample collection, storage, transportation, and processing to minimize any potential sources of bias or error.
It would be helpful for the authors to include information on quality control measures implemented to ensure the reliability and reproducibility of the results. This could involve detailing measures taken to validate assay performance, monitor assay precision and accuracy, and control for potential contamination during sample processing.
It would be advisable for the authors to include information on how these cut-off values of ELISA were validated and whether any external validation methods were employed to ensure the accuracy of the categorization.
The authors may consider performing additional statistical analyses to further strengthen their findings. For example, conducting sensitivity analyses to assess the robustness of results to different model specifications or exploring potential interactions between variables that may influence the outcomes of interest.
In the vaccination trial, it would be beneficial for the authors to incorporate appropriate negative controls to account for any potential non-specific effects of vaccination or adjuvants on the outcomes measured. This could involve including a control group that receives a placebo or mock vaccination to distinguish specific vaccine effects from non-specific immune responses.

Given that the vaccination trial involves the administration of experimental vaccines to broiler breeders, it would be important for the authors to report any adverse events or reactions observed during the trial. This could include monitoring and documenting any clinical signs, mortality rates, or other indicators of vaccine safety and tolerability in both the breeder birds and their offspring.

The conclusions drawn by the authors appear to be consistent with the evidence and arguments presented throughout the study. The conclusions address the main question posed, which is to investigate the potential protective role of maternal antibodies against Campylobacter colonization in broiler chicks.

Based on the provided text, it appears that the references used in the study are appropriate for supporting the findings and interpretations presented.  

Comments on the Quality of English Language

Moderate editing of English language required

Author Response

(The authors gave the same response as above.)

Reviewer 4 Report

Comments and Suggestions for Authors

Dear Authors,

The aim of the manuscript submitted for review is an attempt to evaluate under experimental conditions the protective effect of maternal antibodies in broilers against Campylobacter colonization. It is worth noting that such research has not been carried out in the field so far. The manuscript is well organized, the authors use appropriate scientific language and the obtained results are clearly presented using tables, charts and supplementary materials. The conclusions are written in a concise manner and provide a specific answer to the research goal. I certify that the assessed manuscript presents a high scientific level. I have a few minor comments that were highlighted in the attached copy of the manuscript.

Kind regards

Author Response

(The authors gave the same response as above.)

Round 2

Reviewer 3 Report

Comments and Suggestions for Authors

Accept in present form